# Anterior Pedicled Nasal Flap in Frontal Sinus Drill-Out Patients: A Randomised Controlled Pilot Study

**DOI:** 10.3390/jcm11154329

**Published:** 2022-07-26

**Authors:** Mark Bastianelli, Lucy Huang, Paige Moore, Isma Zafar Iqbal, Charmaine M. Woods, Eng H. Ooi

**Affiliations:** 1Department of Otolaryngology and Head and Neck Surgery, Flinders Medical Centre, Bedford Park 5042, Australia; lucy.huang@sa.gov.au (L.H.); paige.moore.ent@gmail.com (P.M.); ismaiqbal@doctors.org.uk (I.Z.I.); charmaine.woods@flinders.edu.au (C.M.W.); eooi.entsurgery@gmail.com (E.H.O.); 2College of Medicine and Public Health, Flinders University, Bedford Park 5042, Australia

**Keywords:** endoscopic modified Lothrop, Draf III, mucosal flap

## Abstract

**Background:** The endoscopic modified Lothrop procedure (EMLP) is a common procedure performed in patients with frontal sinus pathology. While performing this procedure, large segments of bone are exposed, which may lead to the promotion of frontal sinus neo-ostium stenosis. Here we examine the peri-operative differences in time to achieve healing in patients where a mucosal flap is used to cover the exposed bone on one side of the neo-ostium. **Design:** A randomised pilot study with 12 patients undergoing EMLP surgery participated in this study. **Methods:** Patients were randomised to undergo a mucosal flap on either the left or right side of the neo-ostium. Prior to surgery, patients completed a SNOT-22 and smell identification test. Patients were reviewed until the neo-ostium had healed on both sides. Once healing had occurred, a post-operative SNOT-22 score and smell identification test were recorded. **Results:** Average time to healing for the frontal sinus neo-ostium was 4.7 vs. 4.2 (*p* = 0.3) on the flap vs. non-flap side, respectively. There was an average 24.4 point (range: −75 to +9) decrease in SNOT-22 scores post-surgery. The post-operative USPIT score demonstrated an average increase of 6.6 points (range −13 to +27). **Conclusion:** We did not detect significant differences in peri-operative time toward healing in neo-ostiums where a single flap is utilised. Further studies are needed to determine whether the usage of a single neo-ostium flap affords any benefit over no flap on either ostium. SNOT-22 and UPSIT scores improved post-surgery.

## 1. Introduction and Objectives

### 1.1. Scientific Background

The endoscopic modified Lothrop procedure (EMLP) (also known as Draf III or frontal sinus drill out) is an accepted technique for approaching the frontal sinus in patients with recalcitrant frontal sinus disease or other sinonasal pathology [1,2]. The goal of this operation is to gain access to the frontal sinus and to maximally widen the frontal drainage pathway [3,4]. In order to achieve this, the superior aspect of the septum, frontal beak, frontal sinus floor and intersinus septum are drilled and thus removed [5]. In so doing, this creates a common drainage pathway for both the right and left frontal sinus. However, drilling to remove these structures creates large areas of exposed bone laterally and superiorly, which heals by secondary intention and can promote neo-osteogenesis and contribute to post-operative stenosis of the frontal sinus neo-ostium [6]. A previous meta-analysis by Anderson et al. demonstrated that among 612 patients who underwent an EMLP procedure, the restenosis rate was 14%, with 80% of those patients with restenosis subsequently requiring revision surgery [7].

Multiple studies have previously examined predictors of stenosis as well as techniques to minimise restenosis [5,7,8]. A variety of techniques, including mucosal flaps and free mucosal grafts, have previously been described as techniques to diminish bone exposure and reduce post-operative stenosis. A recent study by Wang et al. demonstrated that patients undergoing EMLP, where flaps or free mucosal grafts were utilised, had a significantly increased anterior-posterior and lateral neo-ostium diameter at their 12-month follow-up compared to a control cohort where no graft or flaps were used [5]. This suggests that techniques used to reduce bony exposure reduce post-operative restenosis. However, no studies have illustrated the post-operative differences in time required for healing between the sides of the neo-ostium in patients where only one flap is used to cover half of the frontal sinus neo-ostium.

### 1.2. Study Objectives

The primary aim of this randomised controlled pilot study was to compare the time required to achieve post-operative healing in patients undergoing EMLP where an anteriorly-based lateral nasal wall flap was used compared to a control side without a flap. As a secondary objective, we evaluated sinonasal outcomes, including smell identification and SNOT22 scores.

## 2. Materials and Methods

### 2.1. Trial Design and Allocation Ratio

The 2010 consort statement for reporting pilot or feasibility studies was followed when appropriate [9]. This study is a randomised matched-pair design pilot study. Patients undergoing EMLP were invited to participate in this study. At the time of surgery, the patient was randomised to undergo an anterior pedicled flap on the right or the left side. The contralateral side was not covered with a pedicled flap or mucosal graft. As such, the allocation ratio for this trial was 1:1 as each individual patient was their own control.

### 2.2. Participant Eligibility

This study was approved by Southern Adelaide Clinical Human Research Ethics Committee (SAC HREC 293.17). Informed consent was obtained from all subjects involved in the study. Inclusion criteria were any patients undergoing an EMLP. Exclusion criteria included any patient who would be unable to follow up post-operatively for routine post-operative reviews.

### 2.3. Setting of Participant Recruitment

Patients were recruited from a rhinology clinic at a tertiary care teaching hospital as well as from Flinders Private Hospital rhinology clinic, both of which are located in Adelaide, Australia. Patients who met the study eligibility criteria were identified as potential candidates for the study. Informed consent was obtained from patients wishing to participate in the trial.

### 2.4. Randomisation and Intervention

Computer-generated randomisation was determined pre-operatively to determine the side that would have a flap and the other side would not have a flap. Sealed envelopes designating which side the anteriorly-based flap should be used were prepared at the commencement of the trial. On the day of surgery, the sealed envelope would be given to the operating surgeon to open. The flap side was recorded in the patient’s data collection sheet as well as their operative report.

### 2.5. Surgical Technique

The surgical technique has previously been described [5]. After the superior septectomy is performed, a nasal flap is raised from the lateral nasal wall (Figure 1). Once this has been raised, the nasal flap is displaced on the ipsilateral side superiorly and anteriorly toward the nasal vestibule so as to avoid injury while drilling the frontal sinus floor (Figure 2). At the completion of the operation, the nasal flap is replaced to cover the exposed bone laterally.

### 2.6. Study Outcomes

Prior to surgery, baseline sinonasal outcomes were recorded, including the Sinonasal Outcome test-22 (SNOT22) [10] and The University of Pennsylvania Smell Identification Test™ [UPSIT] [11].

Patients were followed post-operatively every two weeks (standard clinic protocol) until adequate healing had occurred. At each post-operative appointment, the clinician endoscopically examined the patient after sufficient debridement, and recorded the endoscopic exam for subsequent analysis. Debridement was performed by the rhinology fellow or senior consultant. Based on the endoscopic examination, it was determined whether remnant bony exposure was visible in the frontal sinus neo-ostium or whether healing had occurred. Healing was defined as mucosalisation and absence of exposed bone. Differences between each side of the frontal sinus neo-ostium were recorded.

Post-operative reviews continued at two-week intervals until mucosalisation had occurred on both sides of the frontal neo-ostium. Once mucosalisation had occurred bilaterally, post-operative SNOT-22 and smell identification tests were performed. The results of these tests were compared to pre-operative scores collected on the day of their surgery.

### 2.7. Sample Size

This was a randomised control pilot trial to determine the effect size on time required to achieve post-operative healing and mucosalisation in the presence vs. the absence of a nasal mucosal flap. As this was a pilot trial, no sample size was calculated. The trial was concluded after 12 patients.

### 2.8. Statistical Analysis

Time to healing data were not normally distributed (Shapiro–Wilk); therefore, non-parametric tests were used (Related-samples Wilcoxon signed-rank test).

## 3. Results

### 3.1. Study Participants

Twelve participants were recruited for this study, five female and seven male (Table 1). The median age was 54.8, and age ranged from 23–72 years of age. All patients had undergone previous endoscopic sinus surgery. In 10 cases, the underlying indication for surgery were patients with chronic rhinosinusitis (CRS), defined as refractory sinonasal symptoms for greater than 8–12 weeks with endoscopic or radiologic evidence of sinonasal polyposis or inflammation [12,13]. In addition, one patient had an expansile frontal sinus mucocele that had failed primary surgical management, and one patient with primary ciliary dyskinesia had multiple previous sinonasal surgeries.

### 3.2. Recruitment

The first participant was recruited in September 2018, with the last patient recruited in February 2020. The randomisation allocation was six patients with right-sided flaps and six patients with left-sided flaps.

### 3.3. Result Outcomes

The median healing time to mucosalisation of exposed bone was not statistically different between the non-flap side vs. flap side (4.0, range 2.0–6.0 vs. 4.0, range 2.0–6.0) (*p* > 0.05; Table 1).

Eleven of the twelve patients completed both the pre-operative and post-operative SNOT-22 questionnaire. One patient did not complete the pre-operative SNOT-22 questionnaire. The average pre-operative SNOT-22 score was 50.4 (range 18–81). The post-operative SNOT-22 score was recorded when healing of the frontal neo-ostium had completed. The average post-operative SNOT22 score was 23.9 (range 2–72). Overall, there was a decrease in SNOT-22 scores for our patient population on average by −24.4 points (range −75 to +9, *p* > 0.5). Three patients reported a similar or slight increase in sinonasal symptoms post-operatively.

Smell testing was completed pre-operatively and again post-operatively when the neostium achieved healing on both sides; 8 out of the 12 patients completed both pre-operative and post-operative smell tests. Four patients failed to answer the pre-operative smell test as instructed or declined to complete the post-operative smell test. The average pre-operative UPSIT score was 18.7 (range 8–30). The average post-operative UPSIT score was 28.1 (range 17–30). Overall, an increase in post-operative UPSIT score was observed, with an average increase of 6.6 points (range −13 to +27, *p* > 0.5). Two patients had a decreased post-operative UPSIT score.

Post-hoc chart review demonstrated that patients were followed for an average of 17.8 weeks after surgery (range 6–37 weeks). Of the 12 patients who underwent EMLP, 3 had stenosis develop within 6 months of surgery and required revision EMLP. In all three cases, stenosis was demonstrated bilaterally. There was no difference appreciated between the flap or non-flap side. No other significant complications were recorded in the remainder of our patient population.

## 4. Discussion

Restenosis of the frontal sinus neo-ostium is the most common aetiology for surgical failure in patients undergoing EMLP [14]. While other studies have illustrated that the use of nasal mucosal flaps and free mucosal grafts improved surgical outcomes, our pilot trial did not demonstrate a significant difference in healing time post-operatively between patients with a mucosal flap and those without [1,5,15]. It is conceivable that the follow-up of these patients at two-week intervals was too far apart to detect differences in healing time, and therefore, differences may not have been captured in our study design.

The focus of this study was on the peri-operative healing time after EMLP. Post-hoc chart review, however, demonstrated that three patients had stenosis develop within 6 months of surgery and required revision EMLP. This rate of stenosis is similar to previously published rates of stenosis requiring revision surgery [16]. However, long-term data was not available for the majority of our patient population due to clinic closure and logistical issues encountered in 2020 from the COVID pandemic. Future studies may be able to re-capture this patient population to obtain adequate longer-term data.

This study examined post-operative outcomes in patients with EMLP, where the patient is their own control. This is an important difference from previous study designs, as an alternative theory that could be deduced from this trial is that the post-operative healing time benefited from having at least one flap on a single side. This would be clinically significant and informative to the surgeon as these flaps can sometimes be damaged during surgery or injured while drilling the frontal and nasal bone. As such, if this occurred during a case, it is postulated that having one side with a mucosal flap can still help improve post-operative healing and possibly reduce the risk of stenosis. Further studies with longer-term follow-up would be needed to test this hypothesis, however.

A two-week interval between post-operative assessments was used as this is the interval typically used in the peri-operative period to review patients undergoing EMLP at this centre. For the purpose of this study, we wanted to maintain routine post-operative clinical visits for patients in this pilot study. However, to further evaluate the peri-operative differences in healing post-EMLP, it would be worthwhile for future studies to use a shorter time interval.

Further evaluation in the peri-operative period in this population would be beneficial. Wormald et al. have previously demonstrated the long-term outcomes of patients undergoing EMLP and that frontal nasal ostium has been observed to stabilise 12 months post-surgery [17]. In addition, previous studies have histologically demonstrated that neo-osteogenesis is found in animal models 6 months post-surgery [6]. However, to date, no study has followed the immediate peri-operative differences in healing in patients undergoing EMLP with versus without mucosal flaps. Understanding the normal peri-operative time toward healing can better guide clinical decisions as to the frequency and length of follow-up needed in the immediate post-operative period. Previous reports have shown that in the post-operative period, after sinus surgery, nasal debridement is associated with improved ostium patency and a lower rate of intranasal scarring [18]. As such, one hypothesis that could be deduced is that earlier mucosalisation of exposed bone may lead to less crusting, less need for debridement in the peri-operative period and possibly improved rates of longer-term patency.

Several further limitations to this study have been identified. Surgeons were not blinded post-operatively as to the sidedness of the nasal flap. This introduces the possibility of confirmation bias; we recommend in the next randomised trial that reviewers be blinded to the side of the flap. Furthermore, patients were followed for a short duration post-surgery, limiting the evaluation of the long-term benefits of utilising a mucosal flap. Future studies should formally follow patients for longer follow-up periods to delineate the long-term outcomes of single-sided nasal flaps. A larger sample size is required to adequately power any future RCT. The data from this pilot study was utilised in an a priori power calculation for planning a prospective study. The study would require a sample size of 24 (power 0.80 at α = 0.05) to provide a 1-week improvement in healing. This would require a larger volume of patients meeting the criteria to undergo EMLP. As such, it may be worthwhile for future RCTs to consider a multi-centre trial in order to achieve the necessary patient participation to adequately power a future RCT within a reasonable time period. Finally, there are several factors that may impact post-operative healing (diabetes, smoking, etc.); these factors were not controlled for or standardly recorded as part of our data collection. Future studies should aim to better account for these potential confounding variables.

There is evidence that would suggest that the use of nasal flaps during EMLP can help to prevent surgical failure and restenosis. Our study demonstrated that mucosalisation was achieved in all patients within 6 weeks regardless of whether a flap was placed within the neo-ostium. This leaves the possibility that a single-sided mucosal flap may have provided at least some benefit in post-operative healing for both sides of the neo-ostium. Similar to previous studies, the majority of our patients had improved SNOT-22 scores and that the use of nasal flaps overall did not interfere with post-operative smell outcomes [5]. Further studies are needed to elucidate the peri-operative healing in patients post-EMLP.

## 5. Conclusions

This pilot randomised control trial demonstrated that mucosalisation was achieved within 6 weeks in patients undergoing ELMP with single-sided anteriorly-based lateral nasal wall flap. A future larger multi-site RCT with the proposed amendments are required to examine wound healing and neo-ostium stenosis with the use of a single-sided flap compared to healing by secondary intention.

## Figures and Tables

**Figure 1 jcm-11-04329-f001:**
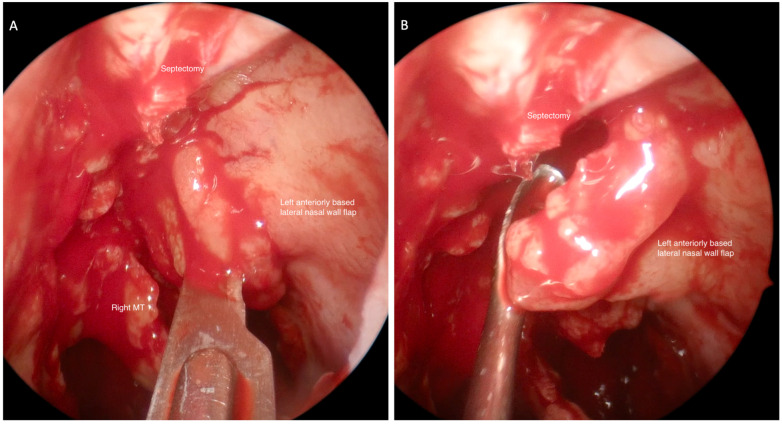
Endoscopic view through right nasal cavity septectomy. (**A**) Raising of a left anteriorly-based lateral nasal wall flap. (**B**) Flap advanced forward and placed into the nasal vestibule to avoid injury during drilling of the frontal sinus ostium. MT—middle turbinate.

**Figure 2 jcm-11-04329-f002:**
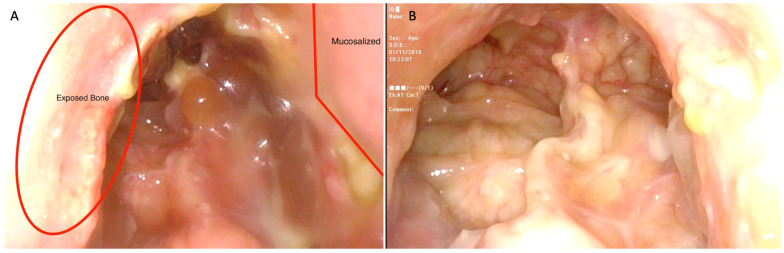
Endoscopic post-EMLP view of frontal neo-ostium. (**A**) Four weeks post-EMLP with left-sided anteriorly-based lateral nasal wall flap. On the right, exposed bone is demonstrated. (**B**) Six weeks post-EMLP frontal neo-ostium demonstrating common drainage pathway for left and right frontal sinus.

**Table 1 jcm-11-04329-t001:** Healing time (weeks) when comparing flap side to non-flap side.

Participant Number	Flap Side	Time to Healing (Weeks)	
		Non-Flap Side	Flap Side
1	right	6	4
2	left	2	2
3	left	2	2
4	Right	6	2
5	right	2	2
6	left	4	6
7	right	6	4
8	left	4	4
9	right	6	6
10	left	6	6
11	right	4	4
12	left	4	4
**Average**		**4.7**	**4.2**
**Median**		**4**	**4**

## Data Availability

The datasets generated and/or analysed during the current study are not publicly available due to patient privacy concerns but are available from the corresponding author on reasonable request.

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
