# Peer review of "Anterior Pedicled Nasal Flap in Frontal Sinus Drill-Out Patients: A Randomised Controlled Pilot Study"

_jcm, 2022, doi:10.3390/jcm11154329_

Round 1
Reviewer 1 Report
It is quite difficult to understand ...
Author had better to show how to make lateral wall flap.
because once pedicled flap can be covered on the bony surface, epithelization and patency both are very good definately.
If not, the procedure must be in sufficient.
Author Response
Response to Review #1
Author had better to show how to make lateral wall flap. Because once pedicled flap can be covered on the bony surface, epithelization and patency both are very good definately. If not, the procedure must be in sufficient.
We have updated the manuscript with some revisions. Thank you for pointing this out, the reference we had cited describing this procedure was incorrectly noted in the body of the manuscript. Instead of reference 7 it should read reference 5. Please also see Figure 1 and figure 2 for some additional detail regarding raising the anteriorly pedicled flap on the lateral nasal wall.
Wang YP, Shen PH, Hsieh LC, Wormald PJ. Free mucosal grafts and anterior pedicled flaps to prevent ostium restenosis after endoscopic modified Lothrop (frontal drillout) procedure: a randomized, controlled study. Int Forum Allergy Rhinol. 2019. doi:10.1002/alr.22416
Sincerely

Reviewer 2 Report
EMLP was performed on 12 patients. The patient was randomized by the side of the nose to have the flap.
There were some minor issues:
1. The author state the design as factorial design may not true. The factorial design needs 2 or more intervention. I suggested to change to randomised matched-pair design.
2. Worth mentioning if there were any blinding process.
3. How frequent and duration the authors follow-up the patient?
4. Did the patients experience any complications or adverse effects?
5. On the post-hoc analysis, please add the information on 3 stenosis patients. The side of stenosis and is it statistically significant?
Author Response
Dear Editor in Chief
It is with great pleasure that I resubmit to you a revised version of our manuscript “Anterior Pedicled Nasal Flap in Frontal Sinus Drill-Out Patients: A Randomised Controlled Pilot Study" Thank you for giving me the opportunity to revise and resubmit this manuscript. We appreciate the time and detail you and the reviewers have provided. I look forward to working with you and the reviewers to move this manuscript closer to publication in the Journal of Clinical Medicine –10th anniversary of JCM– latest advances in the Field of Otolaryngology and Head and Neck Surgery
Response to Review # 2
The author state the design as factorial design may not true. The factorial design needs 2 or more intervention. I suggested to change to randomised matched-pair design.
We agree with this statement. Changes to the manuscript have been made accordingly
- Worth mentioning if there were any blinding process.
Changes to the manuscript have been made. Blinding was not carried out. Patients were randomised to which side the flap would be performed. A sealed envelope was provided to the surgeon on the day of surgery outlining the side of surgery which the flap would be raised. In the post-operative period the fellow or senior rhinologist was not blinded to which side the procedure was performed as this was outlined within the operative report. In the discussion section of our manuscript we have identified this as a potential bias and outlined this as one area of improvement for future study.
- How frequent and duration the authors follow-up the patient?
Patients were followed at two week intervals (standard protocol) until mucosalization had occurred. The study concluded once mucosalization had occurred. However, post-hoc chart review demonstrated that patients were followed on average for 17.8 weeks. We include this in section 3.3. Follow up was hindered by the onset of the covid pandemic. Once mucosalization and healing has occurred we normally follow these patietns quarterly until 12 monhts post-operatively. One of the limitations we mention in our manuscript is lack of long term follow up data partly hindered as this study was concluded at the onset of the Sars-CoV-2 pandemic.
- Did the patients experience any complications or adverse effects?
Please see manuscript changes section 3.3 results outcome
- On the post-hoc analysis, please add the information on 3 stenosis patients. The side of stenosis and is it statistically significant?
Please see manuscript changes section 3.3 results outcome
Sincerely,
Round 2
Reviewer 1 Report
This lateral flap can not cover the hole area of exposed bone.
So It make us mislead to the result of the advantages of the pedicled flap.
Author Response
Dear Editor in Chief
It is with great pleasure that I resubmit to you a revised version of our manuscript “Anterior Pedicled Nasal Flap in Frontal Sinus Drill-Out Patients: A Randomised Controlled Pilot Study" Thank you for giving me the opportunity to revise and resubmit this manuscript. We appreciate the time and detail you and the reviewers have provided. I look forward to working with you and the reviewers to move this manuscript closer to publication in the Journal of Clinical Medicine –10th anniversary of JCM– latest advances in the Field of Otolaryngology and Head and Neck Surgery
Response to Review #1
“This lateral flap can not cover the hole area of exposed bone. So It make us mislead to the result of the advantages of the pedicled flap”
We thank you again for your comments and helping us prepare our manuscript for publication. We have made some adjustments to the wording in the introduction and submit to you again for consideration. The purpose of this study was to examine the peri-operative differences between a neo-ostium which heals by secondary intention versus one that is covered by an anteriorly pedicled mucosal flap. The impact of pedicled free flaps and specifically the anterior pedicled flap has been established previously by Wang et al. However, what is not clear is whether the use of mucosal flaps expedites the time toward healing/mucosalization. To assess this clinical question we designed a randomized pilot study to evaluate peri-operative differences in time toward healing where the patient was their own control. We acknowledge our flap did not cover the entire field of exposed bone however this was not our intention as each patient was their own control. Also, it should be noted that our results do not report a particular advantage in using the mucosal flaps. Indeed, in our study, time to mucosalization/healing was not statistically different between the flap and non-flap sides. In the discussion of our manuscript for this pilot study we identify several changes and improvements that could be used as a framework for readers designing a future study to further evaluate this clinical question. We hope this clarifies your concerns and would be happy to address any other questions or feedback you may have to help us finalize our manuscript for publication.
Sincerely
Mark Bastianelli.
Citation
Wang YP, Shen PH, Hsieh LC, Wormald PJ. Free mucosal grafts and anterior pedicled flaps to prevent ostium restenosis after endoscopic modified Lothrop (frontal drillout) procedure: a randomized, controlled study. Int Forum Allergy Rhinol. 2019. doi:10.1002/alr.22416